# Instructional illustrations in children's learning between normative and realism: An evaluation study

Rommel Mahmoud AlAli[1]*, Ali Ahmad Al-Barakat[2]

1 The National Research Center for Giftedness and Creativity, King Faisal University, Hofuf, Saudi Arabia,
2 Department of Education, University of Sharjah, Sharjah, United Arab Emirates

* ralali@kfu.edu.sa

**Data Availability Statement:** The authors declare that all other data supporting the findings of this study are available within the article.

**Funding:** This work was supported by the Deanship of Scientific Research, King Faisal

## Abstract

Many studies indicate the importance of including the instructional illustrations (pictures, drawings, concrete objects . . .etc.) in childhood education learning materials and employing them in a way that suits the psychological and cognitive levels of young children. In this context, the current study aimed to develop a list of standards to be considered and adopted in designing instructional illustrations, and to reveal the perceptions of childhood teachers about the extent to which these standards are considered in instructional illustrations used in children's learning materials. The participants were childhood education teachers in the Jordanian region of Irbid, who were randomly selected. Two hundred thirty-four teachers completed the questionnaire online. The scale consisted of a total of 34 items distributed over four dimensions. The results showed that the scores of teachers' estimation about employing design standards in the instructional illustrations used in childhood education came at low levels, ranging from average to low, and did not reach high ratings. The study also revealed that there is an impact attributed to teaching experience on teachers' perceptions about the extent to which these standards are employed in instructional illustrations, while there is no impact of gender, academic qualification, or the classes taught by the teachers.

## 1. Introduction

The main objective of the childhood curriculum is to prepare children for life and to combine the experiences and psycho-intellectual abilities that they have been trained in in solving problems so that they can perform their vital activities to the fullest in various situations [1–3]. The educational authorities responsible for the curricula plan them according to the psychological characteristics of the children to achieve the maximum possible degree of growth for their various abilities and aptitudes. Therefore, there is a need for a curriculum in which there are multiple and continuous experiences in which children interact, a variety of educational situations, and opportunities for comprehensive child growth, and the modification of their behavior to the desired educational goals. In addition to integrating thinking skills in the academic content

University, Saudi Arabia [grant number GRANT3643]. The funders had no role in study design, data collection and analysis, decision to publish, or preparation of the manuscript.

**Competing interests:** The authors have declared that no competing interests exist.

and using various appropriate instructional illustrations, the curriculum must also consider the children's tendencies, needs, attitudes, problems, abilities, and aptitudes [4–6].

Children's learning in early grades depends on the nature of the curriculum, which provides them with various learning opportunities that contribute to their full personal development. This leads to their upbringing in a manner consistent with the social philosophy of society. For an effective childhood curriculum, all its components must be effective and meaningful, including instructional illustrations, which play an important role in achieving the desired educational goals, as the use of illustrations is an integral part of the process of achieving any educational learning goal. Therefore, educational institutions are concerned with illustrations in implementing teaching and learning situations to assist the teacher to carry out her/his teaching roles to the fullest, which in turn helps the child to reach the correct understanding and awareness of knowledge in a sound manner, as instructional illustrations facilitate understanding so that abstract things become clear and perceptible to the child [7–10].

The effective contribution of illustrations in its crucial role is to enable children to symbolize abstract knowledge by linking it to memory, as children can, through concrete images, form various symbols and mental images of the objects they learn. In addition, illustrations help children to comprehend different knowledge and information, and store them in long-term memory. Moreover, another role of the illustrations is to develop and improve the children's ability to assimilate sensory perception, which leads to the survival of the experience associated with the illustrations with all its details alive for a longer period in human memory [11–14]. There is a positive effect of sensory games on the children's achievement rate, which helps them to think and increase their learning [15–17].

As for the importance of using instructional illustrations in teaching, they develop and strengthen sensory perception, since pictures, drawings and figures clarify written and audible words for children, provide a tangible material basis for abstract knowledge, train students on organized scientific thinking, develop the ability of children to solve educational problems that they encounter, enable children to classify and distinguish things, replace direct experience, develop children' ability to be accurate, critical, observant and motivated to learn [8, 18].

Due to the importance of illustrations, the Ministry of Education in Jordan has established centres for learning resources and illustrations in each directorate of education in all different regions [19, 20]. This interest came considering the modern educational view that focuses on the child-based learning process, and that the illustrations are a source of learning. This entails that the educational content should be more practical and less ambiguous [21–23]. Based on this view, it has become difficult to separate setting goals and defining appropriate illustrations to achieve those goals, as both are in continuous interaction, as the goal defines the instructional illustrations, while the illustrations are considered as educational aids and effective tools to achieve the goal [2, 19, 24, 25].

There are several classifications of teaching illustrations: perception through the senses that are directly affected and classified into auditory, visual, and audio-visual. Classification according to their function: means of displaying information for the child, materials that are projected onto the screen, such as slides, films, computer software, and transparencies, and materials that do not display lightly, such as stereoscopic images and static images such as photographs, hand drawings and stereoscopic images, and diagrams such as illustrations, sequential drawings, maps and graphs, and educational boards such as blackboards of all kinds and lint boards, magnetic boards, pocket boards, and illustration boards of various types such as hierarchical relationships boards, time boards, assets and branches boards, models, samples, posters, and games [8, 24, 26]. Finally, the classification that divides instructional illustrations according to their effectiveness: passive vectors that do not require the child to actively interact with the educational material, such as television programs, and positive vectors that require

the child to be active in her/his responses and interactions with the educational material, such as television programs and computer-assisted learning [8, 27].

Proceeding from the fact that instructional illustrations are a source of gain in the learning experience, the child becomes more vibrant and active when his learning is related to life and special experiences that reinforce each other [7]. Dale classified instructional illustrations in the cone of experiences according to their impact on the child into three main groups, which included learning with direct and indirect practical experiences, representation, models, samples, trips, exhibitions, museums, field visits, learning with pictorial alternative experiences that include images, animations, stills, and sound recordings, learning through abstract experiences that include verbal symbols and written visual symbols [8, 28].

Employing instructional illustrations occupies a superior place in childhood education, because children do not perceive abstract objects, but rather need to relate objects to their sensory meanings. It is worth noting that the more the senses are involved in the teaching and learning process, the more effective and rapid the learning process will be. Thus, the contemplator of children's learning materials finds that they are full of pictures that contribute to the process of understanding the reading material and help to realize the intended meaning without verbal language, as well as excite children and motivate them to learn [29–31].

Al-Barakat study emphasizes the importance of instructional illustrations, which pointed to the great need for instructional illustrations in the reading book used in teaching first-grade students in Jordan to develop their language skills [32, 33]. The inclusion of instructional illustrations in the learning materials is an indispensable requirement for children in childhood education because they facilitate the occurrence of meaningful learning that leads to understanding and acquiring knowledge, develops the level of the child's mind and thinking, makes her/him perform activities alone, and gives her/him full understanding and accuracy of the lesson [16, 34]. The importance of instructional illustrations also lies in their deep impact on the elements of the educational process (teacher, child, educational material). In addition, educational technology will bring the modern school out of the framework of stereotypes into a new world characterized by scientific and technological achievements [35].

From a psychological point of view, focusing on the use of instructional illustrations in children's learning is one of the main factors that help them learn because the intellectual development of children in the first grades, according to Piaget's theory of cognitive development, is at the beginning of the stage of concrete operations that require supporting children's learning through illustrations and moving away from abstractions. The use of instructional illustrations represents a sensory experience that becomes part of the cognitive structures necessary for thinking in children. Therefore, Brunner's theory confirmed that the child's knowledge-building is related to the extent to which knowledge and information are represented in pictorial ways. The child cannot perceive knowledge in the absence of representation of the abstract object. Therefore, Gagnier considers instructional illustrations as aids for the child to understand the meanings of things while processing the content [16, 36].

Based on the foregoing, instructional illustrations have proven their outstanding role in children's learning. Al-Barakat's study revealed that instructional illustrations are highly effective in developing children's language skills in terms of deriving the main idea from reading lessons, creating class discussion, carrying out tasks related to expression, and helping children interpret the meanings of words and realizing the meaning of the singular and plural in language exercises, in addition, to helping the teacher evaluate the performance of children [32]. Levin and Anglin concluded that learning the theoretical course becomes easy by using educational graphics and illustrations, which in turn lead to multiple functions, including representation, interpretation, transfer, and decoration [37].

Instructional illustrations provide the child with additional information that may not be known to him, which contributes to the formation and development of cognitive structures. Therefore, linking the educational text to an educational image makes the learning process meaningful. Words that are characterized by abstraction can lead to creating ambiguity in the process of acquiring knowledge unless they are combined with the educational picture. Instructional illustrations are very important in improving children's ability to learn and in helping teachers to organize learning. In addition, illustrations help the child absorb, remember and retain what s/he has learned [24, 38].

The results of studies and research in the field of instructional illustrations indicated that the failure to design illustrations effectively is one of the most important factors affecting the young children learning. In this regard, Al-Barakat conducted that the main difficulties in children's learning are due to the lack of clarity in pictures in learning environments, and its inability to consider the individual differences between children, and its irrelevance to children's daily life [32]. The study by Talafha revealed that the design of illustrations and activities as one of the areas of designing social and national education books for the tenth grade was weak, and there is a scarcity of explanatory aids such as tables, statistics, and graphs [39].

The study by Wahiba & Rabiha concluded the importance of illustrations in enriching the course, knowing the individual differences between children, developing thinking in a way of solving problems, developing intelligence, stimulating discussion and interaction, and finally, it is an effective way to attract children's interest [40]. The study by Mahmoud showed that most of the pictures and drawings used in learning process play a positive role in remembering and understanding [41]. Shaheen's study indicated that pictures help in the growth of academic achievement in science and help children to form a positive attitude towards preserving the environment [42]. The study by Al-Ro'oud showed the impact attributed to the picture puzzles method on patterns of social interaction, and the acquisition of chemical concepts [43]. While the study by Abdel Nabi (2002) showed the effectiveness of using picture puzzles in developing picture reading skills and increasing academic achievement in science. Asqoul's study showed varying percentages that the standards for analysis suffer from shortcomings [44]. The study of Mahmoud showed that the use of illustrations helps to develop different thinking processes, observation, description, interpretation, prediction, spatio-temporal relationships, counting, and inference [41]. The study of Abdel-Jalil & Abdel-Wahhab found a positive effect of graphs on achievement and the survival of the effect in science and geography, and the formation of a positive trend [45].

Through the analysis of preceding research [53–55] the significance of integrating instructional illustrations within the educational-learning process becomes evident. This is particularly crucial for young children in their early developmental stages, as their capacity to form meanings, grasp linguistic nuances, and acquire learning encounters is largely reliant on the utilization of instructional illustrations. Their learning is fundamentally grounded in tangible experiences. Consequently, prior investigations have substantiated the criticality of incorporating proficient explanations within educational materials, such as school textbooks, to effectively attain learning objectives for children. Educators emphasize that the adept incorporation of instructional illustrations is intricately tied to their skillful and fitting integration.

Therefore, the present study stands out due to its exploration of educators' evaluations pertaining to educational and instructional illustrations, a facet of paramount significance in the advancement and revitalization of early childhood education. Unveiling the authentic essence of visual aids within children's literature by apprehending the perspectives and evaluations of early childhood educators constitutes a foundational concern in enhancing the caliber of these visual materials. This enhancement aims to empower teachers in effectively leveraging such resources within children's learning milieus. To attain a comprehensive panorama of

educators' appraisals, the current investigation endeavors to discern differentiations in evaluations based on variables like gender, educational attainment, and teaching tenure. This facet assumes great import in the pursuit of educational enhancement. Undoubtedly, the adept crafting of visual aids holds the pivotal key to facilitating children's knowledge acquisition, meaning construction, and semantic development.

Considering the preceding information, the present study stands out due to its approach towards children's books, taking into consideration the principles of effective instructional illustration design. This is particularly significant given the prevailing national focus in Jordan on developing textbooks for all children during their early years. Despite the commendable efforts made towards this objective, there remains a disparity between theoretical principles and practical application. Consequently, this study aims to bridge this gap by addressing the lack of understanding regarding male and female teachers' perspectives on the extent to which instructional illustrations comply with children's learning needs in the Jordanian context.

## 1.1. Study statement

Instructional illustrations occupy a distinct role in the implementation of educational learning situations. The effective investigation of this role depends primarily on its quality [46]. As a result of this great importance, the educational literature has emphasized giving the greatest attention to the design of instructional illustrations that are directly relied upon in the education and learning of children. Instructional illustrations address the most important senses, which is sight, so their quality ensures that information is communicated and understood directly, taking into account the learner's needs and abilities [47].

The study investigated in the lack of consideration of design standards when designing instructional illustrations used in children education. This is confirmed by Al-Barakat's study, which indicated that the poor quality of instructional illustrations design negatively affected the development of language skills of young children in schools [32]. This educational reality called researchers to ask the following question: What are the design standards that should be considered in the design of instructional illustrations used in childhood education? Accordingly, the study attempts to answer the following questions:

1. What is the extent to which the instructional illustrations design standards are employed in those illustrations used in teaching children in schools?

2. Are there statistically significant differences in the estimates of the participants about employing design standards when designing the instructional illustrations used in teaching children according to gender, academic qualification, and teaching experience?

## 1.2. Significance of the study

This study reveals the early learning grade teachers' perceptions about the standards of designing instructional illustrations for those grades. It provides a tool that illustration designers can use in preparing illustrations, especially after opportunities in the information technology era are always available to teachers of the early learning grades to design pictures, graphics, and shapes to use in teaching children. Therefore, the availability of this tool will help teachers to design the illustrations effectively. It also gives a comprehensive vision for those responsible for the educational process about the effectiveness of the instructional illustrations used in teaching children, especially since the Jordanian Ministry of Education is currently going through the phase of the educational reform project for the knowledge economy, especially the focus on the curricula.

### 1.3. Definition of terms

**Childhood teacher:** any teacher who teaches children from kindergarten to the third primary grade. The teacher is responsible for teaching all subjects to the children of these classes.

**Design standards:** are the indicators or semantics that must be considered during the process of preparing, designing, producing, implementing, and presenting instructional illustrations for instructional purposes.

**Instructional illustrations:** are the pictures, figures, graphics, and illustrations that were included in the learning materials for the early grades, as well as the pictures and graphics provided by the Ministry of Education for those grades, whether on paper or electronic format, for display through computer programs.

## 2. Methodology

### 2.1. Approach

The descriptive analytical approach was used due to its suitability to achieve the study objectives.

### 2.2. Participants

The study encompassed all teachers of childhood education in public schools within Irbid Governorate, Jordan, during the second semester of the academic year 2021/2022. Irbid Governorate comprises six Directorates of Education. The objective was to investigate the presence and adherence to standards that should be considered and endorsed in the design of instructional illustrations in a more comprehensive manner. To determine the study sample, a random selection of public schools was made from each of the six departments. The final sample size for this study included 234 male and female teachers.

### 2.3. Procedures

Upon formulating the initial study perception, a comprehensive review of the relevant scientific literature pertaining to the research topic was conducted. Subsequently, a study tool was developed and subjected to rigorous evaluation. To ensure the credibility, validity, and reliability of the study tool, necessary approvals were sought, including the ethical clearance from the Scientific Research Ethics Committee at King Faisal University and the required permissions for implementation (Ref. No. KFU-REC-2023-MAR-EA000479). Upon submission of a formal letter from the university to the Ministry of Education in Jordan, requesting the implementation of specific tools in schools, the Ministry has granted approval for the application. The data collection phase commenced following the aforementioned procedures. A random selection of teachers responsible for teaching the first three grades in schools affiliated with the Directorates of Education in Irbid Governorate constituted the study sample. The developed tool was then distributed among the identified study sample. Subsequently, the tool was collected from the participants, resulting in the acquisition of responses from a total of 234 male and female teachers, yielding a return rate of 95%. Finally, the collected data was meticulously entered into the computer database, and suitable statistical software was employed to conduct the data analysis, ensuring the utilization of appropriate analytical techniques.

### 2.4. Instrument

This study adopted the questionnaire as the main tool for data collection. It was designed by reviewing the literature and previous studies [11–14, 32, 33, 40, 47]. The instrument was used to reveal teachers' perceptions about employing the design standards for instructional illustrations

used in children's learning. It consisted of four dimensions, namely, content validity (11 items), learning knowledge and teaching experiences (7 items), relevance to learning outcomes (8 items), and child needs (8 items). The final form of the instrument consisted of 34 items (S1 Appendix).

The validity and reliability of the instrument were verified which involved expert validation, pilot test, and data analysis using Rasch Measurement Model. Rasch measurement model analysis is a powerful tool for evaluating construct validity. The instrument was validated for construct and content validity by 11 experts from Saudi Universities. In light of their suggestions and opinions, some items were modified, omitted, and added in line with the objectives of the study. After correcting the instrument, the instrument was piloted with 25 respondents. The data were analyzed to measure the validity and reliability using the Rasch Model based on item polarity analysis (point-measure correlation PTMEA) or consistency of the items and these values lies between 0.2 and 1. Second, Infit and misfit, MNSQ value of infit and outfit should lies between 0.4 and 1.5, and standardized fit statistic (Zstd) values should range between -2 and 2. Third, dimensionality aspects, the dimensionality criterion should be more than 40%, and the unexplained variance in the first contrast is less than 15. Fourth, Item and persons separation, the criterion for accepting reliability exceeds 0.50, and acceptable separation should be more than 2 [48, 49].

Table 1 below show dimensionality data results, it is compatible with calibration measurement analysis. The results are in line with the dimensionality analysis because the raw variance is explained by measures greater than 40%, and the unexplained variance in the 1st contrast is less than 15. Therefore, dimensionality data results are appropriate according to the Rasch model.

The validity of the instrument was measured using MNSQ values for the infit, and the results showed that the instrument had an appropriate degree of validity. Instrument validity scores according to MNSQ values fall within the safe limits, which should lie be-tween 0.4 and 1.5. It is consistent with the item polarity analysis according to PTMEA values, whose value should be between 0.2 and 1. It has a suitable standardized fit statistic (Zstd) value, which should be between -2 and 2, as shown in Table 2 below.

The reliability of the instrument was measured using person reliability and Item re-liability of the instrument. The results of the study revealed that the instrument has an appropriate degree of items' reliability of the instrument, as shown in Table 3.

The scale of five categories was used for instructional illustrations instrument that comprised 1 = Unimportant, 2 = Slightly Important, 3 = Moderately Important, 4 = Important, and 5 = Very Important. Table 4 and Fig 1 showed a summary of the category structure on a scale gradation and size structure of the intersection of instructional illustrations. The column arrangement observation (observed count) showed the respondents' answers to the ranking scale. As shown in Table 4, the most frequent answer was the scale of respondents ranking 3

**Table 1. Item dimensionality of instructional illustrations instrument.**

| | Empirical | | Modeled | |
|---|---|---|---|---|
| Total raw variance in observations | 47.8 | 100.0% | | 100.0% |
| Raw variance explained by measures | 13.8 | 47.9% | | 28.1% |
| Raw variance explained by persons | 3.9 | 8.2% | | 8.0% |
| Raw Variance explained by items | 9.9 | 20.7% | | 20.2% |
| Raw unexplained variance (total) | 34.0 | 71.1% | 100.0% | 71.9% |
| Unexplained variance in 1st contrast | 4.2 | 8.8% | 12.4% | |
| Unexplained variance in 2nd contrast | 2.9 | 6.1% | 8.6% | |
| Unexplained variance in 3rd contrast | 2.5 | 5.2% | 7.4% | |
| Unexplained variance in 4th contrast | 2.3 | 4.7% | 6.6% | |
| Unexplained variance in 5th contrast | 2.0 | 4.2% | 5.9% | |

**Table 2. Item fit analysis for instructional illustrations instrument.**

| items | Measure | Model S.E | Infit | | outfit | | Pt-measure | |
|---|---|---|---|---|---|---|---|---|
| | | | MNSQ | ZSTD | MNSQ | ZSTD | CORR | EXP |
| CN3 | 0.06 | 0.07 | 1.31 | 1.1 | 1.40 | 1.8 | 0.43 | 0.46 |
| CN4 | 0.30 | 0.08 | 1.19 | 1.8 | 1.27 | 1.7 | 0.50 | 0.45 |
| E7 | 0.07 | 0.08 | 1.21 | 1.9 | 1.24 | 1.4 | 0.45 | 0.44 |
| L06 | 0.06 | 0.07 | 1.24 | 1.6 | 1.21 | 1.2 | 0.48 | 0.45 |
| L05 | 0.31 | 0.07 | 1.21 | 1.9 | 1.23 | 1.3 | 0.49 | 0.45 |
| E2 | 0.19 | 0.08 | 1.15 | 1.5 | 1.10 | 1.0 | 0.42 | 0.44 |
| CV5 | 0.08 | 0.07 | 1.12 | 1.4 | 1.14 | 1.6 | 0.40 | 0.46 |
| CN1 | 0.07 | 0.07 | 1.14 | 1.4 | 1.14 | 1.4 | 0.46 | 0.46 |
| CV4 | 0.13 | 0.07 | 1.10 | 1.0 | 1.13 | 1.3 | 0.40 | 0.46 |
| L07 | 0.60 | 0.08 | 1.12 | 1.3 | 1.12 | 1.4 | 0.43 | 0.43 |
| E3 | 0.16 | 0.07 | 1.11 | 1.2 | 1.12 | 1.3 | 0.49 | 0.46 |
| CN2 | 0.16 | 0.08 | 1.10 | 1.0 | 1.10 | 1.1 | 0.47 | 0.44 |
| CV7 | 0.74 | 0.08 | 1.08 | 1.0 | 1.09 | 1.1 | 0.45 | 0.44 |
| CV3 | 0.12 | 0.08 | 1.07 | 0.8 | 1.08 | 1.0 | 0.46 | 0.43 |
| CV6 | 0.05 | 0.09 | 1.07 | 0.9 | 1.08 | 0.9 | 0.43 | 0.41 |
| E1 | 0.85 | 0.08 | 1.05 | 0.6 | 1.01 | 0.2 | 0.51 | 0.44 |
| E6 | 0.25 | 0.07 | 0.97 | 0.3 | 1.03 | 0.4 | 0.57 | 0.45 |
| L03 | 0.02 | 0.07 | 1.02 | 0.3 | 1.03 | 0.4 | 0.56 | 0.47 |
| CN7 | 0.04 | 0.07 | 1.02 | 0.2 | 1.00 | 0.0 | 0.44 | 0.46 |
| L02 | 0.07 | 0.06 | 0.94 | 0.8 | 0.92 | 0.9 | 0.55 | 0.52 |
| CV2 | 0.06 | 0.07 | 0.88 | 1.4 | 0.93 | 0.7 | 0.55 | 0.47 |
| CN6 | 0.18 | 0.07 | 0.92 | 1.0 | 0.90 | 1.2 | 0.57 | 0.51 |
| CV1 | 0.50 | 0.06 | 0.91 | 1.1 | 0.88 | 1.4 | 0.60 | 0.55 |
| CN8 | 0.18 | 0.07 | 0.88 | 1.3 | 0.87 | 1.5 | 0.57 | 0.48 |
| L01 | 0.30 | 0.06 | 0.84 | 1.8 | 0.88 | 1.5 | 0.58 | 0.51 |
| CV10 | 0.05 | 0.07 | 0.87 | 1.3 | 0.87 | 1.3 | 0.57 | 0.44 |
| L04 | 0.10 | 0.08 | 0.86 | 1.5 | 0.86 | 1.5 | 0.57 | 0.44 |
| CV9 | 0.10 | 0.06 | 0.85 | 1.7 | 0.84 | 1.0 | 0.61 | 0.51 |
| E5 | 0.51 | 0.06 | 0.84 | 1.6 | 0.81 | 1.4 | 0.62 | 0.51 |
| CN5 | 0.33 | 0.07 | 0.84 | 1.5 | 0.82 | 1.8 | 0.63 | 0.52 |
| L08 | 0.45 | 0.07 | 0.83 | 1.1 | 0.83 | 1.1 | 0.63 | 0.52 |
| CV8 | 0.64 | 0.07 | 0.80 | 1.6 | 0.80 | 1.6 | 0.63 | 0.49 |
| CV11 | 0.08 | 0.08 | 0.80 | 1.7 | 0.79 | 1.4 | 0.63 | 0.46 |
| E4 | 0.44 | 0.07 | 0.71 | 1.9 | 0.70 | 1.5 | 0.71 | 0.52 |

which was 79 (34%). The next grading scale, that respondents selected, was scale 4 out of 55 (24%). Scale 5 had 42 (18%) respondents while the least grading scale of least was scale 1 with 34 (15%) respondents and scale of 2 of 24 (10) respondents. The observed averages showed the pattern of respondents. A fairly normal pattern is expected with a systematic instrument from negative to positive. As illustrated in Table 4.

## 3. Findings

To answer the first study question, what is the extent to which the instructional illustrations design standards are employed in those illustrations used in children's learning materials? The means and standard deviation were extracted for each item as shown in Table 5.

**Table 3. Person and item separation and reliability for instructional illustrations instrument.**

|  | Score | Count | Measure | Error | Infit | | Outfit | |
|---|---|---|---|---|---|---|---|---|
|  |  |  |  |  | MNSQ | ZSTD | MNSQ | ZSTD |
| Mean | 115.4 | 34.0 | 0.41 | 0.19 | 0.99 | 0.3 | 1.01 | 0.2 |
| S.D | 16.8 | 0.0 | 0.64 | 0.02 | 0.50 | 2.1 | 0.52 | 2.1 |
| Real rmse | 0.21 |  |  |  |  |  |  |  |
| Adj. sd | 0.60 |  |  |  |  |  |  |  |
| Separation | 2.89 |  |  |  |  |  |  |  |
| Person reliability | 0.89 |  |  |  |  |  |  |  |
| Mean | 794.4 | 234.0 | 0.00 | 0.07 | 1.00 | 0.1 | 1.01 | 0.0 |
| S.D | 62.0 | 0.0 | 0.33 | 0.01 | 0.15 | 1.7 | 0.16 | 1.8 |
| Real rmse | 0.08 |  |  |  |  |  |  |  |
| Adj. sd | 0.32 |  |  |  |  |  |  |  |
| Separation | 4.19 |  |  |  |  |  |  |  |
| Item reliability | 0.95 |  |  |  |  |  |  |  |

Table 5 shows that the means in the first dimension, learning outcomes, ranged be-tween 2.89 and 3.73, the means of the second dimension, content validity ranged be-tween 2.85 and 3.58, the means of the third dimension, teaching experiences ranged between 2.93 and 3.78, and lastly the mean of the fourth dimension, child needs ranged between 3.06 and 3.61. While the mean of overall instructional illustrations Instrument was 3.32 with medium degree.

To answer the second question, T- Test and one-way analysis of variance were used. Table 6 below shows the results of T-Test for the employing design standards in the instruc-tional illustrations used in teaching children for the difference in instrument dimensions due to gender.

Table 6 shows that the value of (t = 0.077) for whole dimensions indicated that there were no statistically significant difference between the means, where the significant level was more than (0.05). In other word, there were no statistically significant differences be-tween the responses of the sample about employing design standards in the instructional illustrations used in teaching children in the dimensions of the instrument due to gender.

Table 7 below shows the results of one-way analysis of variance for employing de-sign stan-dards in the instructional illustrations used in teaching children for differences in the dimen-sions of the instrument due to academic qualification, teaching experience, and the grades they teach.

Table 7 shows that there were no statistically significant differences between the re-sponses of the sample about employing design standards in the instructional illustrations used in teach-ing children in the dimensions of the instrument due to academic qualification, as the signifi-cant level was more than 0.05. Additionally, Table 7 clarifies that there were statistically

**Table 4. Calibration scaling analysis of instructional illustrations instrument.**

| Category Label | Observed Count % | | Observed Average | Sample Expect | Infit MNSQ | Outfit MNSQ | Structure Calibration | Category Measure |
|---|---|---|---|---|---|---|---|---|
| 1 | 34 | 15 | -0.16 | 0.08 | 0.91 | 0.95 | Non | (1.62) |
| 2 | 24 | 10 | 0.04 | 0.12 | 0.53 | 0.46 | 0.07 | 0.57 |
| 3 | 79 | 34 | 0.22 | 0.30 | 0.54 | 0.46 | 1.28 | 0.19 |
| 4 | 55 | 24 | 0.97 | 0.56 | 0.63 | 0.94 | 0.48 | 1.08 |
| 5 | 42 | 18 | 0.75 | 0.98 | 1.02 | 1.04 | 0.72 | (2.49) |

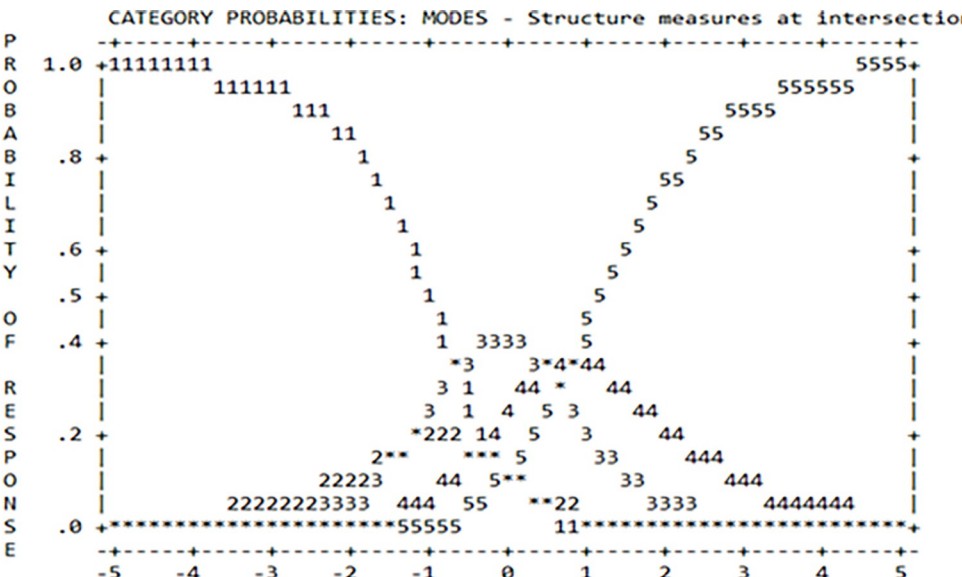

**Fig 1. The summary of the category structure of instructional illustrations instrument.**

significant differences at the level of significance (0.05) in the responses of the sample about employing design standards in the instructional illustrations used in children's learning in the dimensions of the instrument due to teaching experience. To determine the sources of the differences, Tukey's test was used for the post-comparisons was used.

Table 8 shows that there are statistically significant differences between the responses of the sample about employing design standards in the instructional illustrations used in teaching children due to teaching experience. Moreover, these differences are in favor of "less than 5 years".

## 4. Discussion

This study aimed to develop a list of standards for designing instructional illustrations and applying them to teachers of primary grades to reveal their perceptions about employing these standards in instructional illustrations used in teaching children to those grades. This was discussed by answering some questions, and discussing the results related to these questions.

### 4.1. Results related to the first question

Table 2 shows that the scores of estimates about employing design standards in the instructional illustrations ranged between the arithmetic averages of 2.85 and 3.78. It came at an average level. Accordingly, it can be inferred through the analysis of arithmetic averages that there are weaknesses in employing design standards for instructional illustrations adopted in teaching children, and this can be determined as follows:

Although the instructional illustrations are related to the educational objectives to a moderate degree, in addition to being designed to address the main concepts contained in the content, they did not contribute to the development of emotional products or outcomes and did not allow the child to carry out practical activities as required. This is what the study sample showed through the two items that achieved the lowest arithmetic averages. This reveals that the instructional illustrations do not focus to a distinct degree on the emotional outcomes. Respondents also indicated that the explanations were not appropriately designed for the child

**Table 5. The rank, means, and standard deviation for the teachers' estimated scores of the early learning grades about employing design standards in the instructional illustrations.**

| Rank | | N | Mean | Std. Deviation |
|---|---|---|---|---|
| 1 | LO7 | 234 | 3.7308 | .90785 |
| 2 | LO5 | 234 | 3.7094 | .99406 |
| 3 | LO3 | 234 | 3.5855 | 1.13620 |
| 4 | LO6 | 234 | 3.4615 | 1.01100 |
| 5 | LO4 | 234 | 3.2350 | .85937 |
| 6 | LO2 | 234 | 3.2051 | 1.19396 |
| 7 | LO1 | 234 | 3.2009 | 1.26620 |
| 8 | LO8 | 234 | 2.8889 | 1.16662 |
| | Learning Outcomes | 234 | 3.3496 | .67151 |
| 1 | CV7 | 234 | 3.5829 | .87466 |
| 2 | CV6 | 234 | 3.5795 | .79957 |
| 3 | CV3 | 234 | 3.5325 | .86017 |
| 4 | CV4 | 234 | 3.5026 | 1.04417 |
| 5 | CV10 | 234 | 3.4637 | 1.01135 |
| 6 | CV5 | 234 | 3.3573 | 1.05243 |
| 7 | CV2 | 234 | 3.3316 | 1.05920 |
| 8 | CV9 | 234 | 3.2269 | 1.18940 |
| 9 | CV11 | 234 | 3.1594 | .89444 |
| 10 | CV8 | 234 | 2.9162 | 1.08160 |
| 11 | CV1 | 234 | 2.8504 | 1.32290 |
| | Content Validity | 234 | 3.3185 | .60971 |
| 1 | TE1 | 234 | 3.7821 | .93544 |
| 2 | TE2 | 234 | 3.6453 | .95288 |
| 3 | TE3 | 234 | 3.5513 | 1.01467 |
| 4 | TE7 | 234 | 3.4573 | .90320 |
| 5 | TE6 | 234 | 3.3462 | 1.01271 |
| 6 | TE5 | 234 | 2.9573 | 1.20723 |
| 7 | TE4 | 234 | 2.9274 | 1.16044 |
| | Teaching Experiences | 234 | 3.3551 | .66976 |
| 1 | LN4 | 234 | 3.6094 | .97223 |
| 2 | LN2 | 234 | 3.5513 | .92157 |
| 3 | LN7 | 234 | 3.4957 | 1.04511 |
| 4 | LN3 | 234 | 3.4658 | 1.05681 |
| 5 | LN8 | 234 | 3.4402 | 1.02262 |
| 6 | LN1 | 234 | 3.4188 | .99991 |
| 7 | LN6 | 234 | 3.2034 | 1.14872 |
| 8 | LN5 | 234 | 3.0641 | 1.17564 |
| | Child Needs | 234 | 3.2886 | .69844 |
| | Overall instructional illustrations Instrument | 234 | 3.3280 | .57075 |

to carry out practical activities, which does not lead to the instructional illustrations achieving the effective role they should play in the development of children's learning in various respects. Perhaps the reason for this is due to the belief of those responsible for primary education and the designers of illustrations that the child should take a general idea through the illustrations while ignoring the partial things. If this belief exists, it represents one of the factors affecting the child's learning. Accordingly, the effective effect of educational pictures was not employed

**Table 6. Results of T-Test for differences between means according to gender.**

| Variables and Dimensions | | | No. | Mean | Std. Deviation | T-Value | Sig. |
|---|---|---|---|---|---|---|---|
| Gender | Learning Outcomes | Male | 87 | 3.2839 | .66958 | .209 | .251 |
| | | Female | 147 | 3.3884 | .67189 | | |
| | Content Validity | Male | 87 | 3.4057 | .66496 | .011 | .821 |
| | | Female | 147 | 3.4245 | .57679 | | |
| | Experiences | Male | 87 | 3.2943 | .65154 | .140 | .281 |
| | | Female | 147 | 3.3912 | .67994 | | |
| | Children Needs | Male | 87 | 3.3713 | .74629 | .386 | .169 |
| | | Female | 147 | 3.5014 | .66641 | | |
| | Overall average | Male | 87 | 3.3356 | .60174 | .077 | .441 |
| | | Female | 147 | 3.3952 | .55250 | | |

to achieve children's learning outcomes, especially since instructional illustrations are employed to help the child perceive the educational material. Reid et al., referred to [32], say that instructional illustrations carry a message for the child that cannot be realized through words.

The results of the study show that the focus in the design of instructional illustrations did not reach a very high or high degree of estimation, but rather came at an average level. This result reveals the absence of an important criterion considering which instructional illustrations should be designed. This result contradicts the educational goals expected to be achieved by children, especially since the educational process seeks to build the child and develop the mental abilities that make him able to deal and interact positively with the requirements of life [27]. It also does not focus on the child as the focus of learning effectively, also, it does not link

**Table 7. Results of analysis of variance of differences between the means of responses of sample.**

| Variance Source | | | Sum of Squares | df | Mean Square | F | Sig. |
|---|---|---|---|---|---|---|---|
| Academic qualification | Learning Outcomes | Between Groups | .305 | 3 | .102 | .223 | .880 |
| | | Within Groups | 104.760 | 230 | .455 | | |
| | Content Validity | Between Groups | 1.203 | 3 | .401 | 1.080 | .359 |
| | | Within Groups | 85.415 | 230 | .371 | | |
| | Experiences | Between Groups | .348 | 3 | .116 | .256 | .857 |
| | | Within Groups | 104.171 | 230 | .453 | | |
| | Children Needs | Between Groups | .454 | 3 | .151 | .308 | .820 |
| | | Within Groups | 113.208 | 230 | .492 | | |
| | Whole Dimensions | Between Groups | 1.104 | 3 | .368 | 1.132 | .337 |
| | | Within Groups | 74.796 | 230 | .325 | | |
| Teaching Experience | Learning Outcomes | Between Groups | .231 | 2 | .116 | .255 | .045 |
| | | Within Groups | 104.834 | 231 | .454 | | |
| | Content Validity | Between Groups | 1.068 | 2 | .534 | 1.441 | .039 |
| | | Within Groups | 85.551 | 231 | .370 | | |
| | Experiences | Between Groups | 4.006 | 2 | 2.003 | 4.603 | .011 |
| | | Within Groups | 100.513 | 231 | .435 | | |
| | Children Needs | Between Groups | 1.563 | 2 | .782 | 1.611 | .002 |
| | | Within Groups | 112.099 | 231 | .485 | | |
| | Whole Dimensions | Between Groups | 1.148 | 2 | .574 | 1.773 | .032 |
| | | Within Groups | 74.753 | 231 | .324 | | |
| | | Within Groups | 75.818 | 230 | .330 | | |

**Table 8. Results of Tukey's test for differences between the lengths of experience of teachers in relation to employing design standards in the instructional illustrations used in teaching children.**

| Mean | (I) Experience | (J) Experience | Mean Difference (I-J) | Std. Error | Sig. |
|---|---|---|---|---|---|
| 3.4356 | Less than 5 years | 5–10 years | .11984 | .14675 | .693 |
| | | more than 10 years | -.02644 | .14341 | .041 |
| 3.2857 | 5–10 years | Less than 5 years | -.11984 | .14675 | .693 |
| | | more than 10 years | -.14629 | .07839 | .151 |
| 3.4020 | more than 10 years | Less than 5 years | .02644 | .14341 | .041 |
| | | 5–10 years | .14629 | .07839 | .151 |

the child's previous experiences with the new, and there is an absence of linking them to reality and various life contexts. This result is consistent with the study by Wahiba & Rabiha [40].

The design of illustrations, which should be based on the pupils' past and life experiences, as well as its ability to address problems of daily life, was not very effective as it came in the middle level. Given the importance of these standards, they should assume a very high or significant degree; because the child in his learning builds knowledge through a process of integration between new knowledge and his previous cognitive structure, whether he learned it in previous classes or it is related to the social reality in which he lives, which allows him to discuss and dialogue with the teacher or with his colleagues [50].

In light of the foregoing, the lack of instructional illustrations to help the child is no longer limited to the extent of their clarity and output in colors that reflect the studied phenomenon, but rather the results of the study revealed that the illustrations did not meet some of these standards, and negatively affected the process of helping the child to extract and build knowledge. These standards can be inferred by taking into account the instructional illustrations of the individual differences between children, and the nature of their mental and psychological development, which did not achieve high scores of estimations. Thus, the lack of attention to these two standards in the design of instructional illustrations significantly affects the knowledge building of the child. Learning experiences may not occur as desired if they do not match the abilities of the children.

Despite the great importance of following the instructional illustrations with accompanying activities, they did not receive high levels of interest. This represents one of the obstacles that affect children's learning because instructional illustrations are a source for practicing classroom and extracurricular activities. This result contradicts previous studies that emphasized the importance of illustrations as a source of classroom activities (Al-Barakat, 2003). Accordingly, children will find it difficult to construct and acquire knowledge through illustrations, as constructive learning confirms that the child builds his knowledge in the light of practicing a series of activities [51].

In light of the preceding discussion, these findings deviate from prior research [52–55] that underscores the pronounced role of instructional illustrations in enhancing the caliber of children's education. Consequently, the outcomes highlighted by the absence of clarifications are poised to exert an adverse impact on a child's psychological disposition, diminishing their curiosity and impeding their motivation to engage with and grasp cognitive subjects. This is particularly significant given that instructional illustrations serve as a potent instrument for shaping children's learning encounters in a captivating and stimulating manner.

## 4.2. Results related to the second question

The results of the t-test showed that there were no statistically significant differences among the responses of the sample according to gender. This indicates that the estimates of the study

sample did not differ according to the gender of the respondent. Perhaps this result confirms that male and female teachers' experiences and knowledge are similar in determining the standards for designing instructional illustrations. It also confirms that male and female teachers hold the same knowledge and use the same standards to judge whether the illustrations achieve effective standards or not.

Table 7 shows that there were no statistically significant differences due to academic qualification. This confirms that the discrepancy in academic qualifications did not affect the teachers' estimates of the instructional illustrations design standards. The absence of statistically significant differences between the childhood education teachers according to their academic qualification could indicate that their academic studies did not differ in the definition of the students with the illustrations' standards. Therefore, a teacher's qualification variance does not cause a variance in grade averages. This may indicate that child-hood education teachers, despite their different academic qualifications, hold converging views on the instructional illustrations they use.

It could be asserted that the absence of statistically significant distinctions in teachers' responses, considering both gender and educational qualifications, may be ascribed to the circumstance that male and female educators possessing diverse academic credentials share a distinct clarity regarding the prerequisites for crafting instructional illustrations. This shared insight has facilitated instructors in effectively conveying their perspectives.

Table 7 shows that there were statistically significant differences in the responses of the sample about employing design standards in the instructional illustrations used in teaching children in the dimensions of the instrument due to teaching experience in favor of inexperienced teachers and the teachers with the less teaching experience. This result is attributed to the fact that teachers with short experiences have more knowledge than those with long experiences, which could indicate that teachers with short experience have been exposed to more experiences in the field of illustration design during pre-service preparation than their counterparts with long experience, especially since teachers with short experience have studied in Jordanian universities according to the plans developed in the light of educational reform.

These findings align with the outcomes of preceding studies [56–58] that underscored the significance of training teachers in the adept utilization of instructional illustrations. Consequently, a substantial proportion of novice educators have participated in training initiatives focused on the effective deployment of instructional illustrations. Moreover, the inclusion of instructional illustration implementation as an integral facet within pre-service teacher professional development programs has been emphasized, congruent with the tenets of the constructive learning model that accentuates the proactive role of the learner. This outcome can likewise be attributed to the context in which individuals with limited teaching experience, particularly those new to the profession, embark upon their initial years of service with zeal, dynamism, and heightened motivation, particularly as they find themselves in a trial phase.

In general, the findings of the study indicate that the instructional illustrations used in childhood education have received great attention. This can be attributed to the existence of a perception among the designers of instructional illustrations that childhood stage represents the beginning of education for young people, which necessitates providing them with illustrations with high levels of interest, especially since their learning is based mainly on semi-tangibles, as modern educational perceptions confirm that children's learning in the early learning grades is necessary to design instructional illustrations according to certain standards that lead to the achievement of effective learning.

## 5. Conclusion, limitations, and future directions

This study was concerned with developing a list of standards to be considered and adopted in the design of instructional illustrations, then revealing the extent of their use in those illustrations used in teaching and learning children, and showing the impact of gender, teaching experience, academic qualification, and the grade taught by the study sample on the scores of their estimation about employing those standards in teaching. The study reached the development of a list of standards for designing instructional illustrations, where all standards obtained relative importance (80%) or more from the point of view of the teachers of the early learning grades. The study also revealed that the scores of their estimation about employing design standards in the instructional illustrations used in teaching children came at low levels, ranging from average to low.

Based on the findings of this investigation, it is deducible that the pragmatic implications of the study underscore the imperative of directing attention towards the principles governing the design of educational and instructional illustrations. These illustrations constitute a pivotal wellspring that empowers educators to assume the role of learning activity architects and facilitators, fostering children's learning through the proficient integration of educational illustrations. Furthermore, insights drawn from the study's results affirm that the adept utilization of educational illustrations yields a substantial impact on shaping children's acquisition of knowledge, conceptual understanding, skills, values, and attitudes. Such adept utilization facilitates active learning experiences and nurtures children's accountability for their own educational journey. This resonates with the contemporary global movement emphasizing the central position of the child in the learning process. This overarching conclusion is realized by nurturing children's engagement with educational illustrations, wherein they engage in diverse scientific thought processes, thereby cultivating a deeper understanding of self and subject matter.

The study has several limitations that should be acknowledged. Firstly, it is constrained by its focus on childhood teachers' perceptions pertaining to the criteria for designing educational illustrations specifically utilized in children's educational materials. The study does not encompass other perspectives or stakeholders involved in the design process. Secondly, the research tool employed in this study is limited to a resolution questionnaire. Alternative methods of data collection, such as interviews or observations, were not incorporated, which may have provided a more comprehensive understanding of the topic. Thirdly, the study's scope is confined to assessing the utilization of the criteria for designing educational clarifications as outlined in the educational materials for childhood education, including those provided by the Ministry of Education in paper or electronic formats. Consequently, the findings cannot be generalized to illustrations found in children's stories or other printed materials beyond the specific context of educational clarifications.

In light of these main results of the study, the study recommends the following: first: adopting the list of design standards developed in this study as a basis for designing the instructional illustrations that teachers produce using computer programs. Second: bene-fit from the list of design standards developed in this study when schools purchase children's pictures, posters, textbooks, stories, and illustrated magazines. Third: utilizing a list of developed design standards to judge the effectiveness of instructional illustrations used in higher grades, and children's stories and picture books sold in public libraries. Fourth: holding workshops for childhood education teachers to train them to produce instructional illustrations in light of the list of approved design standards. Finally, it is worth conducting educational studies that reveal the extent to which authors of children's stories and their comic books employ the list of design standards of the instructional illustrations developed in this study.

Further Research on Standards: Conduct additional research to refine and expand the existing standards for designing educational clarifications in the first three grades. This can involve exploring specific subject areas, pedagogical approaches, and learning outcomes to ensure comprehensive coverage. Longitudinal Studies: Undertake longitudinal studies to examine the long-term impact of employing the standards of designing educational clarifications. Assess how their implementation affects students' academic performance, engagement, and overall learning outcomes as they progress through higher grades. Teacher Training and Professional Development: Develop and provide specialized training programs and professional development opportunities for teachers to enhance their understanding and application of the standards in designing educational clarifications. This can include workshops, courses, and mentoring programs that focus on effective instructional strategies, curriculum alignment, and assessment practices. Collaboration and Sharing Best Practices: Foster collaboration among educators, educational institutions, and stakeholders to facilitate the sharing of best practices and successful implementation of the standards. This can be achieved through conferences, symposiums, online platforms, and communities of practice dedicated to the exchange of ideas, resources, and experiences. Technology Integration: Investigate the integration of technology tools and digital resources in the design of educational clarifications to enhance engagement and facilitate personalized learning. Explore how technologies such as interactive multimedia, virtual reality, and gamification can be leveraged to align with the standards and improve students' learning experiences. Continuous Evaluation and Feedback: Establish a system for continuous evaluation and feedback to monitor the effectiveness of implementing the standards. This can involve collecting data on student performance, teacher practices, and student and parent perceptions. Use this feedback to inform improvements and adjustments to the standards and their implementation strategies. Inclusive Education Considerations: Incorporate considerations for inclusive education in the standards of designing educational clarifications. Ensure that the design of clarifications caters to the diverse needs of students, including those with disabilities, English language learners, and students from different cultural backgrounds. Collaboration with Stakeholders: Engage parents, school administrators, policymakers, and other stakeholders in the process of designing and implementing educational clarifications. Seek their input and involvement to foster a shared vision and commitment to effective implementation.

## Supporting information

**S1 Appendix.**
(DOCX)

## Author Contributions

**Methodology:** Rommel Mahmoud AlAli.

**Writing – original draft:** Rommel Mahmoud AlAli, Ali Ahmad Al-Barakat.

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
