## [Decision Letter · Decision Letter 0]

30 Jun 2023

PONE-D-23-03499Instructional Illustrations in Children’s Learning between Normative and Realism: An Evaluation StudyPLOS ONE

Dear Dr. Rommel,

Thank you for submitting your manuscript to PLOS ONE. After careful consideration, we feel that it has merit but does not fully meet PLOS ONE’s publication criteria as it currently stands. Therefore, we invite you to submit a revised version of the manuscript that addresses the points raised during the review process. Please read the reviewers' comments carefully and make changes accordingly.

We look forward to receiving your revised manuscript.

Kind regards,

Anastassia Zabrodskaja, Ph.D.

Academic Editor

PLOS ONE

Journal Requirements:

2. Please provide additional details regarding ethical approval in the body of your manuscript. In the Methods section, please ensure that you have specified the name of the IRB/ethics committee that approved your study.

“Deanship of Scientific Research at King Faisal University, Saudi Arabia Grant..”

“The authors thank the Deanship of Scientific Research at King Faisal Univer-sity, Saudi Arabia for the financial support under Annual research grant number GRANT.”

“Deanship of Scientific Research at King Faisal University, Saudi Arabia Grant..”

7. Please provide additional details regarding participant consent. In the ethics statement in the Methods and online submission information, please ensure that you have specified what type you obtained (for instance, written or verbal, and if verbal, how it was documented and witnessed). If your study included minors, state whether you obtained consent from parents or guardians. If the need for consent was waived by the ethics committee, please include this information.

8. We note that you have referenced (ie. Bewick et al. [5]) which has currently not yet been accepted for publication. Please remove this from your References and amend this to state in the body of your manuscript: (ie “Bewick et al. [Unpublished]”) as detailed online in our guide for authors

Reviewers' comments:

Reviewer's Responses to Questions

**Comments to the Author**

1. Is the manuscript technically sound, and do the data support the conclusions?

Reviewer #1: Yes

Reviewer #2: Yes

2. Has the statistical analysis been performed appropriately and rigorously? 

Reviewer #1: Yes

Reviewer #2: Yes

3. Have the authors made all data underlying the findings in their manuscript fully available?

Reviewer #1: Yes

Reviewer #2: Yes

4. Is the manuscript presented in an intelligible fashion and written in standard English?

Reviewer #1: Yes

Reviewer #2: Yes

5. Review Comments to the Author

Reviewer #1: This is a well-written paper that may have contributions. I identified several issues that need to be addressed by the authors.

1. In the introduction section, it is imperative for the authors to enhance their efforts in effectively emphasizing the existing gaps in the literature. By doing so, they can clearly articulate how their research aims to address and bridge these gaps, thereby contributing to the advancement of knowledge in the field.

2. In the methodology and research design section, it is imperative to provide a comprehensive definition of the population under study as well as an explanation of the sampling technique employed. Additionally, it is essential to justify the appropriateness of the chosen sampling technique and the determined sample size.

3. In order to enhance the quality and depth of the discussion section, it is crucial to incorporate a comprehensive comparison between the current findings and the existing body of literature. By engaging in such a comparative analysis, the research outcomes can be contextualized, evaluated, and potentially contribute to the advancement of knowledge in the field.

4. The authors must emphasize both the practical and theoretical implications of this study to enhance its significance. While the conclusion section briefly touches upon some managerial implications, they are presented in a rather limited fashion. In order to provide a comprehensive understanding of the study's potential real-world applications, it is crucial for the authors to delve deeper into the practical implications. This can be achieved by elaborating on how the findings of the study can be translated into actionable strategies or recommendations for practitioners in relevant fields.

5.Acknowledging research limitations and outlining future directions for further work is a crucial aspect of any scholarly endeavor. However, regrettably, the importance of addressing these elements has been overlooked or neglected in this particular study. Therefore, in order to rectify this oversight, it is imperative to not only recognize the limitations inherent within the research but also to shed light on potential avenues for future investigation.

Reviewer #2: The study is fascinating and relevant to the fields of education, psychology, and human development. However, there are two aspects to review in the presentation of the results:

(1) The empirical framework focuses on studies on the importance of illustrations and images in children's learning rather than on teachers' assessments. It is this second aspect that the study evaluates and, therefore, must be justified in more detail. In this regard, it is striking that the results of the teachers' assessment are extrapolated to the effects of the illustrations on children's learning in the discussion section. Such inferences should not take place directly as they do not share the phenomenon of inquiry. Strictly speaking, this is a study on teachers' assessment, which is important in educational terms. However, more is needed to contribute to studying the effects of illustrations or images on children's learning.

(2) The second hypothesis tested is not justified in the background, so the discussion of the findings is unclear: Why were differences by gender, age, and experience expected? Why are these differences significant for the first objective?

6. PLOS authors have the option to publish the peer review history of their article (what does this mean?). If published, this will include your full peer review and any attached files.

Reviewer #1: No

Reviewer #2: No

---

## [Author Response · Author response to Decision Letter 0]

29 Aug 2023

Dear Reviewers,

I would like to express my sincere gratitude for your diligent efforts in reviewing the manuscript and providing valuable feedback. Your expertise and insightful comments have greatly contributed to the refinement of the work.

I am pleased to inform you that I have addressed all the concerns and suggestions raised during the review process. Enclosed herewith is the "Response to Reviewers" document, which outlines the modifications made in accordance with your recommendations.

Should there be any further modifications or adjustments requested, please do not hesitate to reach out to us. We are committed to ensuring the highest quality of the manuscript and welcome any additional input.

Once again, I appreciate your time, expertise, and invaluable contribution to improving the manuscript.

Sincerely,

---

## [Decision Letter · Decision Letter 1]

1 Sep 2023

Instructional Illustrations in Children’s Learning between Normative and Realism: An Evaluation Study

PONE-D-23-03499R1

Dear Dr. Rommel,

We’re pleased to inform you that your manuscript has been judged scientifically suitable for publication and will be formally accepted for publication once it meets all outstanding technical requirements.

Kind regards,

Anastassia Zabrodskaja, Ph.D.

Academic Editor

PLOS ONE

Additional Editor Comments (optional):

Reviewers' comments:

Reviewer's Responses to Questions

**Comments to the Author**

1. If the authors have adequately addressed your comments raised in a previous round of review and you feel that this manuscript is now acceptable for publication, you may indicate that here to bypass the “Comments to the Author” section, enter your conflict of interest statement in the “Confidential to Editor” section, and submit your "Accept" recommendation.

Reviewer #1: All comments have been addressed

2. Is the manuscript technically sound, and do the data support the conclusions?

Reviewer #1: Yes

3. Has the statistical analysis been performed appropriately and rigorously? 

Reviewer #1: Yes

4. Have the authors made all data underlying the findings in their manuscript fully available?

Reviewer #1: Yes

5. Is the manuscript presented in an intelligible fashion and written in standard English?

Reviewer #1: Yes

6. Review Comments to the Author

Reviewer #1: Thank you for addressing the reviewers' comments. The quality of this submission has increased significantly. I am satisfied with the current version.

7. PLOS authors have the option to publish the peer review history of their article (what does this mean?). If published, this will include your full peer review and any attached files.

Reviewer #1: No

---

## [Editor Report · Acceptance letter]

6 Sep 2023

PONE-D-23-03499R1 

Instructional Illustrations in Children’s Learning between Normative and Realism: An Evaluation Study 

Dear Dr. AlAli:

I'm pleased to inform you that your manuscript has been deemed suitable for publication in PLOS ONE. Congratulations! Your manuscript is now with our production department. 

Kind regards, 

on behalf of

Professor Anastassia Zabrodskaja 

Academic Editor

PLOS ONE